# A Soft Labeling Approach for Fairness-aware Learning Under Partially Annotated Sensitive Attributes

## Abstract

In light of AI's growing ubiquity, concerns about its societal impact have prompted extensive efforts to mitigate different types of bias, often relying on the assumption of complete information regarding individuals' sensitive attributes. In this work, we tackle the problem of algorithmic fairness under partially annotated sensitive attributes. Previous approaches often rely on an attribute classifier as a proxy model to infer "hard" pseudo labels, which are then used to optimize the final model using fairness-aware regularization techniques. In contrast, we propose a novel regularization approach, that leverages the output probability of the attribute classifier as "soft" pseudo labels, derived from the definition of the fairness criteria. Additionally, we study the effect of the uncertainty on the attribute classifier parameters that naturally arise in the case of limited available sensitive attribute annotations. We adopt the Bayesian viewpoint and we propose to optimize our model with respect to the marginal model of the attribute classifier, while our second approach optimizes the fairness objective with respect to each model of the decision maker's belief. To validate our approach, we conduct extensive experiments on Adult and CelebA datasets with tabular and image modalities, respectively. The results of our study highlight the effectiveness of our method as well as the significance of incorporating uncertainty, in improving both utility and fairness compared to a variety of different baselines.

## 1 Introduction

Artificial Intelligence (AI) has rapidly emerged as a powerful and transformative technology that undeniably impacts our everyday lives. Organizations frequently employ AI-powered automated decision-making applications as solutions to complex problems, enhancements to existing solutions, or to minimize human effort across various tasks. Some of the most promising and beneficial applications of AI include Natural Language Processing (NLP) for machine translation (Bahdanau et al., 2015), Computer Vision (CV) in healthcare for diagnosis and treatment (Esteva et al., 2021), and Reinforcement Learning (RL) for self-driving cars (Bojarski et al., 2016), enhancing transportation safety and efficiency.

On the other hand, problems regarding the reliability, safety, and biases of AI systems have raised concerns about the negative impact of AI on our society. Algorithmic fairness has become a focus area, addressing ethical and societal implications associated with potential biases and discrimination arising from the use of AI algorithms, particularly in applications that directly impact people.

The question of fairness is rather philosophical. To determine what is fair, one first has to answer the question of how fairness should formally be defined in a certain setting (Bechavod & Ligett, 2017). There are two main notions of algorithmic fairness: individual and group fairness. The concept of individual fairness (Dwork et al., 2012) states that similar individuals should be treated similarly by the AI system. The notion of group fairness considers the different treatment of the groups and the potential harms of the disadvantaged groups (Hardt et al., 2016), based on statistical parity metrics where different individuals are grouped based on some sensitive attributes such as race or gender. Research efforts are primarily devoted to analyzing different notions of fairness or

developing practical fairness-aware learning algorithms to mitigate the biases of the models while maintaining a desirable level of performance.

Most of the existing methods that try to mitigate biases in the group fairness notation silently assume fully annotated sensitive information of individuals in both the training and evaluation phases. However, in many practical applications, this assumption does not hold, and we often end up with no or partial information about a sensitive attribute, making the learning of fairness-aware models especially difficult. This concept naturally arises in many different scenarios where legal or ethical concerns force the data acquisition procedure to limit the collection of sensitive-related information. A straightforward example is the personal data protection law, i.e., the EU General Data Protection Regulation (GDPR), where the consent of the users is required to store and manipulate sensitive information. Another motivational example involves human annotation, which can be resource-intensive, time-consuming, or ethically sensitive.

In this work, we consider the problem of fairness-aware learning under partly annotated sensitive attributes. More specifically, we consider the case where our dataset $D$ consists of a potentially large part with no annotations about the sensitive information $D_U$ and another part with full annotations about the sensitive attributes $D_L$ that can be significantly smaller, i.e., $|D_U| \gg |D_L|$. Most of the existing methods approach the problem by estimating the sensitive attributes of the unlabeled data using an attribute classifier (often called a proxy model) trained on the labeled data, and then applying any fairness-aware method to mitigate the bias of the model. We argue that the use of these proxy models that produce "hard" sensitive labels can potentially harm the overall fairness performance of the model due to over-fitting incorrect sensitive labels. Similar arguments about confirmation bias are also present in the literature of semi-supervised learning (Arazo et al., 2020) and learning with noisy labels (Tanaka et al., 2018).

Our contribution is to propose a soft labeling approach to learn fairness-aware models when only partial information about sensitive attributes is available. More specifically, our method consists of fair regularization that employs the probability of each sensitive class based on the output of a probabilistic attribute classifier. We provide theoretical justifications for our approach derived from the imposed fairness definition. Leveraging the information from the unlabeled data can intuitively increase the predictive performance of our model, while we experimentally validate that utilizing the information about the sensitive information encoded in the unlabeled data has also benefits regarding fairness. In addition, we study the effect of uncertainty, which naturally arises when the labeled training data $D_L$ is relatively small for estimating missing sensitive information. To tackle this challenge, we introduce two alternative uncertainty formulations for the proposed soft regularization method, drawing inspiration from the work of Dimitrakakis et al. (2019) on Bayesian fairness. To empirically evaluate our approach, we conduct a comprehensive evaluation study using diverse datasets and data modalities, comparing the results of our approach with various existing baselines.

## 2 RELATED WORK

**Fairness under imperfect information:** The concept of fairness under imperfect information can be categorized based on the type of information that is missing. In this work, we study the setting of partial annotation sensitive information where only a fraction of the dataset contains the information of sensitive characteristics, which was first introduced in the work of Jung et al. (2022). Apart from the partial information setting, there is also the extreme case of no sensitive information at all, similar to unsupervised learning (Lahoti et al., 2020; Gupta et al., 2018; Buet-Golfouse & Utyagulov, 2022; Coston et al., 2019). Another interesting setting is the case where there is additional missing information related to task labels. This scenario bears resemblance to semi-supervised learning (SSL), as explored in Zhang et al. (2020); Chzhen et al. (2019); Noroozi et al. (2019).

Next, we delve into the primary methods concerning our setting of partial sensitive attribute information in the literature. A straightforward solution, motivated by the work of Lee et al. (2013) on SSL, is to train a group attribute classifier and assign pseudo labels to the group-unlabeled data as a preprocessing step before applying any bias mitigation technique. The aforementioned work of Jung et al. (2022) uses a similar method that assigns group pseudo labels only when the classifier is sufficiently confident, by tuning the classification threshold. Another interesting approach with the aim of learning fair representation that can be used in various downstream tasks is the work of Zhang et al. (2022) which employs constructive learning in the computer vision domain using highly

accurate generative models. The most relevant to our work is the paper of Chen et al. (2019), which also employs a soft labeling approach but with the aim of assessing model bias rather than learning fairness-aware models.

**Fairness under uncertainty:** The academic community has recently expressed growing interest in the relationship between uncertainty and algorithmic fairness. In this work, we adopt the Bayesian Fairness framework introduced by Dimitrakakis et al. (2019) that studies the effect of parameter uncertainty from a decision-theoretic perspective. More specifically, they proposed a Bayesian-like approach to calculate the imposed fairness criteria with respect to a set of possible world models according to the decision maker's belief, while the follow-up work of Athanasopoulos et al. (2023) proposes a bootstrap method to scaling the framework for the continuous data case. The significance of incorporating model uncertainty was also highlighted in Russell et al. (2017) from a point of view of causal modeling. Additionally, a broader discussion about uncertainty as a form of transparency is presented by Bhatt et al. (2021).

## 3 PRELIMINARIES

In this section, we begin with some preliminaries on algorithmic fairness and the notation used in this work. Without loss of generality, we consider the binary classification setup where we have a dataset of $N$ individuals $D = \{x_i, z_i, y_i\}_{i=1}^N$ with inputs $x \in \mathcal{X} \subseteq \mathcal{R}^d$, sensitive attributes $z \in \mathcal{Z} = \{0, 1\}$ and labels $y \in \mathcal{Y} = \{0, 1\}$. We consider the probabilistic classifier $f : \mathcal{R}^d \to [0, 1]$ that outputs the model belief $s \in [0, 1]$ that an individual $n$ belongs to the positive class i.e. $P(\hat{y} = 1 \mid x)$, where the predicted label $\hat{y} \in \hat{\mathcal{Y}} = \{0, 1\}$ is obtained using a threshold $t \in [0, 1]$ i.e. $\hat{y} = \mathbf{1}_{s \geq t}$ [1].

In terms of group fairness, we want the decisions of our classifier to be fair according to some independence criteria based on sensitive attribute $z$ while maximizing the task utility. Some of the most famous independence criteria are demographic parity (DP) (Feldman et al., 2015) and equalized odds (EO) (Hardt et al., 2016). More specifically, DP states that the model predictions $\hat{y}$ should be independent from the sensitive attribute $z$ i.e. $\hat{y} \perp z$ witch can equivalently be expressed as follows:

$$P(\hat{y} \mid z = \alpha) = P(\hat{y} \mid z = \beta) \ \forall \alpha, \beta \in \mathcal{Z} \tag{1}$$

On the other hand, equal opportunity (EO) states that prediction $\hat{y}$ should be independent of the sensitive attribute $z$ when conditioned to the true outcome $y$ i.e. $\hat{y} \perp z \mid y$ which can equivalently be expressed as:

$$P(\hat{y} \mid z = \alpha, y) = P(\hat{y} \mid z = \beta, y) \ \forall \alpha, \beta \in \mathcal{Z} \tag{2}$$

Regularization methods are usually employed as a scalable approach to mitigate bias while preserving accuracy in models with a high number of parameters. More specifically the final model $f$ can be trained with the stochastic gradient descent (SGD) algorithm to minimize a combined loss that accounts for both utility and fairness:

$$\min_f \ \mathcal{L}_{utility}(f) + \lambda_f \mathcal{L}_{fairness}(f) \tag{3}$$

where $\mathcal{L}_{utility}(f) = \mathbb{E}_{x,y \sim P}[l(x, y; f)]$ denotes the utility loss while, with $l$ typically denoting the cross-entropy loss. In addition $\mathcal{L}_{fairness}$ refers to the loss that tries to mitigate the bias according to the fairness definition. For DP and EO Madras et al. (2018) propose the two following relaxed metrics that are commonly employed in the literature:

$$\mathcal{L}_{fairness}^{DP} = \mid \mathbb{E}_{x \sim P_0} f(x) - \mathbb{E}_{x \sim P_1} f(x) \mid \ , \ \mathcal{L}_{fairness}^{EO} = \sum_{y \in \mathcal{Y}} \mid \mathbb{E}_{x \sim P_0^y} f(x) - \mathbb{E}_{x \sim P_1^y} f(x) \mid \tag{4}$$

where we define $P_a = P(\cdot \mid z = a)$ and $P_a^y = P(\cdot \mid y, z = a)$. The aforementioned metrics are usually estimated by using the empirical deviation of the expectations.

---

[1] $\mathbf{1}_{condition}$ is the indicator function where is equal to 1 if the condition is true, 0 otherwise

# 4 Soft Fair Regularization Under Partial Sensitive Information

## 4.1 Method

In this work, we are interested in the setting where our dataset has partially annotated sensitive information. We introduce a simple and intuitive regularization approach that leverages the output probabilities of each sensitive class obtained from an auxiliary sensitive attribute classifier, denoted as $f_z$, rather than relying on 'hard' pseudo labels. In the current section, we present our soft regularization method for both equalized odds (Eq.2) and demographic parity (Eq.1).

More formally, we assume that our dataset $D$ is partitioned into a labeled dataset $D_L = \{x, z, y\}_{i=1}^{N_L}$ and a group-unlabeled dataset $D_U = \{x, y\}_{i=1}^{N_U}$, with size of $N_L$ and $N_U$ respectively. Our research question is how we can utilize the non-annotated part of the dataset $D_U$ to enhance both the utility and fairness of our final model $f$. Intuitively when $N_U \gg N_L$ the unlabeled part of the dataset $D_U$ contains a lot of information about the joint distribution of $(X, Y)$ that can be leveraged to improve both the utility and fairness.

In order to utilize the unlabeled dataset $D_U$, we express the fairness loss $\mathcal{L}_{fairness}$ of the fair regularization method (Eq.3) as a combined loss that accounts both for the loss of the labeled part $\mathcal{L}_L$, and loss of the unlabeled part $\mathcal{L}_U$ of the dataset, weighted according to a scaling parameter $\lambda_U$.

$$\mathcal{L}_{fairness} = (1 - \lambda_U)\mathcal{L}_L + \lambda_U \mathcal{L}_U \tag{5}$$

The unlabeled part of the dataset has no information about the sensitive attribute $z$ and therefore the regularization criteria of equation 4 cannot be directly applied. We propose the following reformulation of the fairness-aware regularisation criteria:

$$\mathcal{L}_U^{DP} = \sum_{y \in \mathcal{Y}} | \; \mathbb{E}_{x \sim P} \left[ f(x) \frac{p(z = 0 \mid x)}{p(z = 0)} \right] - \mathbb{E}_{x \sim P} \left[ f(x) \frac{p(z = 1 \mid x)}{p(z = 1)} \right] | \tag{6}$$

$$\mathcal{L}_U^{EO} = \sum_{y \in \mathcal{Y}} | \; \mathbb{E}_{x \sim P^y} \left[ f(x) \frac{p(z = 0 \mid x, y)}{p(z = 0 \mid y)} \right] - \mathbb{E}_{x \sim P^y} \left[ f(x) \frac{p(z = 1 \mid x, y)}{p(z = 1 \mid y)} \right] | \tag{7}$$

where we rewrite the expectations using basic probability theorems:

$$\mathbb{E}_{x \sim P_a} [f(x)] = \mathbb{E}_{x \sim P} \left[ f(x) \frac{p(z = a \mid x)}{p(z = a)} \right] \tag{8}$$

$$\mathbb{E}_{x \sim P_a^y} [f(x)] = \mathbb{E}_{x \sim P^y} \left[ f(x) \frac{p(z = a \mid x, y)}{p(z = a \mid y)} \right] \tag{9}$$

In order to calculate our final metrics (Eq.6 and 7), we need to compute the two probabilities: $p(z = a \mid x, y)$ and $p(z = a \mid y)$ or $p(z = a \mid x)$ and $p(z = a)$ according to the desired criteria, using an auxiliary model. The first quantity, $p(z = a \mid x, y)$ or $p(z = a \mid x)$, can be estimated using an attribute classifier $f_z$ that outputs the probability of each sensitive class $z$. The second term, $p(z = a \mid y)$ or $p(z = a)$, can also be estimated using discrete probability models $f_d$. Both models can be trained using the labeled dataset $D_L$. Implementation-wise, the overall proposed algorithm is displayed in Algorithm 1.

Compared to the existing approaches in the literature for handling partially annotated sensitive attributes, our soft regularization method weights each example according to the probability of each sensitive attribute in the empirical deviation metric (Eq.6 and 7). Intuitively, examples with low confidence, where the probabilities of the sensitive attributes are close, contribute less to our soft fair regularization. In contrast, the pseudo-labeling approach assigns a label to each example, thereby non-confident examples influencing the fair regularization. Additionally, methods that adjust the classification threshold to ignore less confident predictions may still exhibit some residual bias in confident predictions, with performance depending on the threshold adjustment.

---

**Algorithm 1** Soft Fair Regularization Under Partial Sensitive Information ( Equalised Odds )

---

**Require:** Group labeled $D_L$ data, Group unlabeled $D_U$ data, total iteration T, scaling algorithm parameters $\lambda$ and $\lambda_f$, learning rate $\eta$ and batch size $b$

Train attribute classifier $f_a$ using $D_L$

Train district probability model $f_d$ using $D_L$

**for** $t$ in $T$ **do**

   *// Sample mini-batches*

   $B_U = \{B_U^y = (x_i, y)_{i=1}^b \sim D_U^y\}_{y \in Y}$

   $B_L = \{B_{a_L}^y = (x_i, z, y)_{i=1}^b \sim D_{z_L}^y\}_{(z,y) \in (Z \times Y)}$

   $B = B_U \bigcup B_L$

   *// Calculate Utility loss //*

   $\mathcal{L}_{utility}(B; f) = \mathbb{E}_{x, y \sim B} [l(x, y; f)]$

   *// Calculate Fair loss //*

   $\mathcal{L}_U^{EO}(B_U; f, f_d, f_a) = \sum_{y \in \mathcal{Y}} | \mathbb{E}_{x \sim B_U^y} \left[ f(x) \frac{f_a(x,y)}{f_d(y)} \right] - \mathbb{E}_{x \sim B_U^y} \left[ f(x) \frac{1 - f_a(x,y)}{1 - f_d(y)} \right] |$    ▷ eq. 7

   $\mathcal{L}_L^{EO}(B_L; f) = \sum_{y \in \mathcal{Y}} | \mathbb{E}_{x \sim B_{0_L}^y} f(x) - \mathbb{E}_{x \sim B_{0_L}^y} f(x) |$    ▷ eq. 4

   $\mathcal{L}_{fairness} = (1 - \lambda_f)\mathcal{L}_L^{EO} + \lambda_f \mathcal{L}_U^{EO}$    ▷ eq. 5

   *// Optimise model f //*

   $\mathcal{L}_{total} = \mathcal{L}_{utility} + \lambda \mathcal{L}_{fairness}$    ▷ Total loss

   $\theta_f^t \leftarrow \theta_f^{t-1} - \eta \nabla \mathcal{L}_{total}$    ▷ SGD

**end for**

---

## 4.2 Uncertainty aware regularization

The current learning setting naturally depends on the uncertainty associated with the auxiliary models $f_z$, $f_d$ (Eq.6 and 7). When the labeled data is significantly smaller in size compared to the unlabeled data, and the former has a small size relative to the difficulty of the task then the attribute classifier $f_z$ and discrete probability model $f_d$ can have a large epistemic uncertainty that may affect the coverage of our final model $f$.

In this section, we discuss how we can incorporate uncertainty into our proposed regularization method, motivated by the work of Dimitrakakis et al. (2019), which proposes a Bayesian viewpoint that explicitly takes into account parameter uncertainty. We focus on the epistemic uncertainty of the auxiliary models $f_z$ and $f_d$, to highlight the effect of uncertainty in the setting of partially sensitive information.

More formally when we train a model we have a belief $\beta \in B$ which expresses our uncertainty about the true world model parameter $\theta^*$ of the joint distribution of our data $(x, z, y) \sim P_{\theta^*}$. The belief can be formed as a probability distribution over a set of parameters $\theta \in \Theta$ that may contain the actual parameter $\theta^*$. In our setting, we have some uncertainty regarding the parameters of the different auxiliary models $f_d$ and $f_z$ that can affect the fairness properties of our final model through our proposed regularization.

If are willing to ignore uncertainty, then we can use the marginal model $p_m = \int_\Theta p_\theta d\beta(\theta)$ calculated from our belief. Then we can define the following fair regularization:

$$\mathcal{L}_{U_m}^{DP} = \sum_{y \in \mathcal{Y}} | \mathbb{E}_{x \sim P} \left[ f(x) \frac{p_m(z = 0 \mid x)}{p_m(z = 0)} \right] - \mathbb{E}_{x \sim P} \left[ f(x) \frac{p_m(z = 1 \mid x)}{p_m(z = 1)} \right] | \tag{10}$$

$$\mathcal{L}_{U_m}^{EO} = \sum_{y \in \mathcal{Y}} | \mathbb{E}_{x \sim P^y} \left[ f(x) \frac{p_m(z = 0 \mid x, y)}{p_m(z = 0 \mid y)} \right] - \mathbb{E}_{x \sim P^y} \left[ f(x) \frac{p_m(z = 1 \mid x, y)}{p_m(z = 1 \mid y)} \right] | \tag{11}$$

If we want to really take uncertainty into account, we can use a Bayesian approach. This measures how fair our model $f$ is with respect to each possible parameter $\theta \in \beta$ in our belief and weights each according to the probability of each model $\beta(\theta)$:

$$\mathcal{L}_{U_\beta}^{DP} = \sum_{y \in \mathcal{Y}} \int_\Theta \mid \mathbb{E}_{x \sim P} \left[ f(x) \frac{p_\theta(z = 0 \mid x)}{p_\theta(z = 0)} \right] - \mathbb{E}_{x \sim P} \left[ f(x) \frac{p_\theta(z = 1 \mid x)}{p_\theta(z = 1)} \right] \mid d\beta(\theta) \qquad (12)$$

$$\mathcal{L}_{U_\beta}^{EO} = \sum_{y \in \mathcal{Y}} \int_\Theta \mid \mathbb{E}_{x \sim P^y} \left[ f(x) \frac{p_\theta(z = 0 \mid x, y)}{p_\theta(z = 0 \mid y)} \right] - \mathbb{E}_{x \sim P^y} \left[ f(x) \frac{p_\theta(z = 1 \mid x, y)}{p_\theta(z = 1 \mid y)} \right] \mid d\beta(\theta) \quad (13)$$

In the Bayesian case, the belief $\beta$ is a posterior formed through the prior and available data. However, even if the prior can be calculated in closed form, our metrics (Eq. 13, 12) can only be approximated by sampling from the posterior to perform Bayesian quadrature. Calculation of the posterior itself must also be approximated in general. If we are using Markov chain Monte Carlo, then posterior samples can be directly obtained. In some cases it is also possible to sample from Bayesian neural networks (MacKay, 1992). A more general method is to use an ensemble of models, in lieu of samples from a posterior distribution. In neural networks, we can elaborate (Gal & Ghahramani, 2016) to obtain a set of networks. More generally, we can use some form of resampling of the original data and fit each different model.

In this work, to form our belief we use the bagging ensemble technique (Breiman, 1996), where ensemble members are neural networks trained on different bootstrap samples of the original training set. To get the marginal model we simply average the output probabilities for each class over the different models. To approximate the Bayesian objective (Eq. 13, 12) we can average the fairness objective over their different models. This allows us to perform stochastic gradient descent by calculating the gradients for each sampled model separately, similarly to Dimitrakakis et al. (2019).

## 5 EXPERIMENTS

In this section, we empirically evaluate the effectiveness of our approach, by comparing the results of our method to the standard approaches on fairness-aware learning under partial information introduced in section 2. More specifically we test our method using two different benchmarks, the Adult and the CelebA datasets with tabular and image modalities, respectively. We provide our code[2] which can be used to reproduce all the presented results.

**Evaluation:** We randomly partition our dataset into train, validation, and test sets with sizes $60\%$, $20\%$, and $20\%$ respectively. We use the training set to train our algorithm and we additionally use early stopping in the validation set to prevent models from over-fitting. To test our approach we compare the trade-off between accuracy and fairness highlighting the benefits of our method. We performed each experiment multiple times reporting the mean and the standard deviation of each metric evaluated on the hold-out test set.

**Setting:** To extensively study the effect of our approach we performed multiple experiments using different proportions for the labeled $D_L$ and group-unlabeled data $D_U$. It is worth mentioning that in every experiment the validation and test set remain the same. Moreover, in each different partition setting, we use the same auxiliary model $f_z$, $f_d$ trained in the labeled part of our dataset $D_L$.

**Baselines:** We compare the results of the following baseline methods listed below for all the different experiments. For each baseline that makes use of an attribute classifies $f_z$ to estimate the sensitive attribute label we also perform the optimization using instead the empirical marginal model $f_{z_m} = \frac{1}{N} \sum_i^N f_{z_i}$ of $N = 16$ different attribute classifiers that were trained using a different bootstrap sample. In addition, we also perform the Bayesian approach of equation 13 for our method as explained in section 4.2.

- **vanilla** (Vanilla) The baseline that is trained using empirical risk minimization on the labeled dataset $D_L$ without any fair regularization terms.
- **reg_gap** (Fairness-aware regularization) This method uses the standard fairness-aware regularization technique (Eq.4) on the labeled dataset $D_L$.

---

[2]to be published

- **pseudo** (Pseudo-Label approach) This assigns sensitive labels at $D_U$ using the attribute classifier $f_z$ as a preprocessing step. During the optimization phase, it applies the regularization (Eq.4) as before. **pseudo_m** is the same, but uses the marginal model to impute the sensitive attribute.

- **clg** (Confidence-based pseudo-label) The approach of Jung et al. (2022) that uses only confident pseudo-labels by adjusting the classification threshold. The **clg_m** variant uses the marginal model.

- **soft_reg** (Soft Regularization) Our soft regularization method , as explained in Section 4. We implement the marginal and Bayesian (Eq.13, 11) approaches explained in the uncertainty-aware section (Section 4.2). The latter variants are named **soft_reg_m, soft_reg_b** respectively.

Each of the aforementioned baselines where trained using the same base network architecture and hyper-parameters provided in the following subsections.

## 5.1 ADULT DATASET

In this section, we present the results of the UCI Adult dataset (Becker & Kohavi, 1996) that contains information about over 40,000 individuals from the 1994 US Census. The classification task is to predict whether a particular person has an income greater than \$50.000. As a sensitive attribute, we consider the binary gender information. We perform multiple experiments using different proportions of labeled and unlabeled datasets.

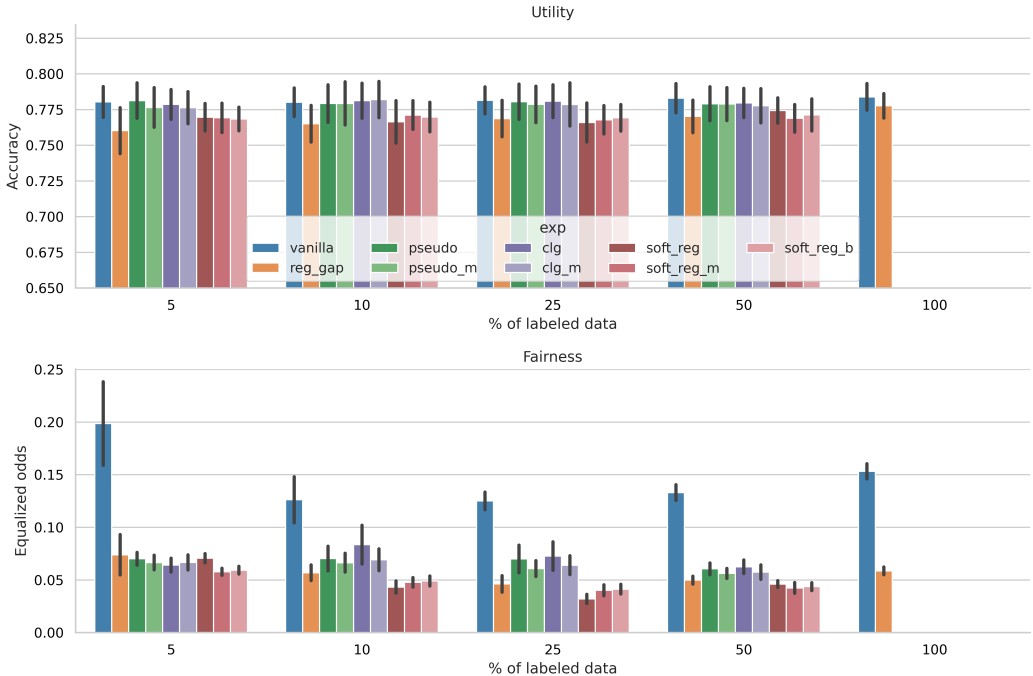

Figure 1: **Results on adult dataset:** The Results on Adult dataset for different methods and percentage of labeled data $D_L$ over 50 runs. We provide the average and the standard deviation for the accuracy and equalized odds evaluated in the hold-out dataset. It's important to recall that regarding Equalized Odds (EO), lower disparities are indicative of better performance.

**Implantation details:** We follow a similar setup to Chuang & Mroueh (2021). More specifically we use simple feed-forward neural networks for both the attribute classifier $f_z$ and the final model $f$. Both models are two-layered neural networks with 512 and 64 units respectively followed by a soft-max layer for the classification. For the first two layers, we use the ReLU (Nair & Hinton,

2010) activation function and l2 regularization with a 0.0001 regularization factor. We train each baseline 50 times using early stopping on the total loss (Eq.3) on the validation set. During the training, we sampled 4 batches with 50 samples in each iteration, one for each pair of labels and sensitive attribute to calculate the empirical version of the different regularization terms according to the methods. Finally, we used a scaling parameter of $\lambda_f = 1.0$ and $\lambda_U = 0.5$ on the equations 3 and 5 respectively.

### 5.1.1 RESULTS

Figure 1 illustrates the results of various baselines under different proportions of labeled and unlabeled datasets. In a broader view, the results validate the general assumption that group unlabeled data is informative for both utility and fairness. More specifically, regarding accuracy, all methods utilizing group unlabeled data perform similarly to each other, while they are relatively close to the $vanilla$ approach and significantly outperform the $reg\_gap$ method. Additionally, we observe that the performance of all baseline methods, in terms of accuracy, remains similar across the different proportions of labeled data, even when compared to the $vanilla$ approach with 100% labeled data.

In terms of fairness, our soft regularization method consistently achieves lower bias across varying proportions of labeled and unlabeled samples. It is noteworthy that our method outperforms the standard regularization method even when 100% of the labeled data is available. Furthermore, methods that account for uncertainty (marginal and Bayesian) appear to result in fairer policies compared to the standard version of the relative method. The complete table corresponding to this figure can be found in Appendix A.

## 5.2 CELEBA DATASET

To further examine the performance and the scaling of our method to high-dimensional tasks we use the CelebA image dataset (Liu et al., 2018). CelebA is a large-scale image dataset with 200.000 images of celebrities' faces extracted from movies and public appearances, where each image is associated with 40 human-annotated labels. As a classification task, we performed two different experiments we performed experiments using two different downstream classification tasks, on the attractiveness and smiling annotation while for the sensitive attribute, we used the binary gender information in both cases. Moreover, due to high computational requirements, we perform experiments only in the case of 5% labeled dataset.

**Implementation details:** Implementation-wise for each baseline and both models $f$ and $f_z$ we use the EfficientNetV2B0 (Tan & Le, 2021) architecture of the convolution neural network along with one hidden layer of 512 units and a final soft-max classification layer. The hidden layer has a ReLU activation function and 0.0001 L2 regularization factor. In addition, we use batch normalization and a dropout layer with a dropout rate of 0.2 before the hidden layer. We initialize the EfficientNetV2B0 architecture with the weights trained on ImageNet. We use early stopping on the total loss measured on the validation set. Due to computational requirements, we use the maximum size of 60 samples per batch equally defined for each sensitive and task label as before. Moreover, we apply horizontal and random flips as well as a random rotation as an augmentation step to further regularize our network. Finally, we use the following scaling parameter of $\lambda_f = 1.0$ and $\lambda_U = 0.5$ on the equations 3 and 5 respectively.

### 5.2.1 RESULTS

In Table 1, we present the results of our experiment applied to the CelebA dataset, using two different downstream tasks. The results clearly demonstrate that our methods outperform the baselines in both experiments. More specifically, we observe that the Bayesian version of our algorithm ($soft\_reg\_b$) achieves superior performance in terms of fairness across the different baselines and both labeled data proportion scenarios. Surprisingly, our method also performs exceptionally well in terms of accuracy, indicating that our uncertainty-aware approach efficiently regularizes our model to achieve a better trade-off between utility and fairness. Another advantage of our approach is that it exhibits significantly less variance in all available metrics, indicating that our regularization helps the model converge consistently in all runs of the algorithm. Moreover, in most cases, the version that incorporates parameter uncertainty in the auxiliary models performs better in terms of fairness compared to other methods. Lastly, it's worth mentioning that the $vanilla$ method for the smiling

| $D_L$ % | Method | task = smiling | | task = attractive | |
|---|---|---|---|---|---|
| | | Accuracy | Equalized Odds | Accuracy | Equalized Odds |
| 100% | vanilla | **0.7914±0.09** | 0.3369±0.290 | **0.6977±0.105** | 0.3876±0.293 |
| | reg_gap | 0.7653±0.14 | **0.2507±0.295** | 0.6777±0.121 | **0.2214±0.292** |
| 5% | vanilla | 0.7880±0.10 | 0.2008±0.129 | 0.6523±0.113 | 0.5886±0.383 |
| | reg_gap | 0.7752±0.10 | 0.3327±0.315 | 0.6923±0.094 | 0.2289±0.082 |
| | pseudo | 0.7631±0.12 | 0.2484±0.256 | 0.7073±0.055 | 0.1969±0.246 |
| | pseudo_m | 0.7441±0.14 | 0.1796±0.212 | 0.6812±0.052 | 0.2965±0.286 |
| | clg | 0.8012±0.10 | 0.2052±0.243 | 0.6802±0.073 | 0.3145±0.362 |
| | clg_m | 0.7318±0.13 | 0.2596±0.269 | 0.7085±0.033 | 0.2192±0.236 |
| | soft_reg | 0.7498±0.10 | 0.2929±0.306 | 0.7386±0.026 | 0.1300±0.206 |
| | soft_reg_m | 0.7574±0.13 | 0.2129±0.221 | **0.7406±0.032** | 0.1315±0.156 |
| | soft_reg_b | **0.8299±0.10** | **0.1230±0.097** | 0.7143±0.079 | **0.0853±0.059** |

Table 1: **Results on CelebA dataset:** The Results on CelebA dataset for different methods and percentage of labeled data $D_L$ over 10 runs. We provide the average and the standard deviation for the accuracy and equalized odds evaluated in the hold-out dataset. It's important to recall that in the context of Equalized Odds (EO), lower disparities are indicative of better performance. It's important to recall that regarding Equalized Odds (EO), lower disparities are indicative of better performance.

task, when applied to just 5% of the training data, surpasses most of the other methods in terms of fairness while still maintaining a high level of accuracy.

## 6 CONCLUSION

In this work, we study the problem of fairness-aware learning under partially sensitive attribute information. We propose a fairness-aware regularization method that makes use of the soft labels of attribute classifiers to alleviate the information of a group-unlabeled dataset. We also consider the effect of the uncertainty of the attribute classifier, which is naturally connecting in the case of the extremely low group labeled data size, proposing two alternative uncertainty-aware versions of our method inspired by the Bayesian fairness framework (Dimitrakakis et al., 2019). To validate our method we perform two experiments on a tabular and high-dimensional image modality, comparing the trade-off between accuracy and fairness. The results validate the assumption that the group unlabeled data is informative for both utility and fairness, while our soft regulation method consistently outperforms all the available baselines. Moreover, we observe that the uncertainty-aware methods outperform the standard single-model approach, especially in the case of a high-dimensional image dataset where we have a greater degree of uncertainty due to the difficulty of the task. We hope that our work will inspire the feather exploration of fairness under incomplete information as well as motivate the integration of uncertainty-aware methods to improve fairness

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

# A    ADULT RESULTS

The following table contains the results on the adult dataset discussed in the experiments section 5.1.1.

| $D_L$ % | Method | Total Loss $\mathcal{L}_{total}$ | Cross Entropy $\mathcal{L}_{utility}$ | Accuracy | Equalized Odds $\mathcal{L}_{fairness}$ |
|---------|--------|------------|---------------|----------|----------------|
| 1.00 | vanilla | 0.580+-0.016 | 0.427+-0.017 | 0.784+-0.009 | 0.153+-0.007 |
|      | reg_gap | 0.504+-0.011 | 0.446+-0.010 | 0.778+-0.009 | 0.059+-0.004 |
| 0.50 | vanilla | 0.563+-0.021 | 0.430+-0.018 | 0.783+-0.010 | 0.133+-0.007 |
|      | reg_gap | 0.501+-0.015 | 0.451+-0.013 | 0.770+-0.011 | 0.050+-0.004 |
|      | pseudo | 0.503+-0.017 | 0.442+-0.016 | 0.779+-0.012 | 0.061+-0.006 |
|      | pseudo_m | 0.504+-0.018 | 0.448+-0.017 | 0.779+-0.012 | 0.056+-0.005 |
|      | soft_reg | 0.495+-0.012 | 0.449+-0.011 | 0.774+-0.009 | 0.046+-0.003 |
|      | soft_reg_m | 0.500+-0.014 | 0.458+-0.012 | 0.769+-0.010 | 0.042+-0.005 |
|      | soft_reg_b | 0.499+-0.014 | 0.455+-0.013 | 0.771+-0.011 | 0.044+-0.004 |
| 0.25 | vanilla | 0.575+-0.024 | 0.449+-0.021 | 0.781+-0.009 | 0.125+-0.008 |
|      | reg_gap | 0.500+-0.020 | 0.453+-0.015 | 0.769+-0.013 | 0.046+-0.008 |
|      | pseudo | 0.520+-0.021 | 0.450+-0.019 | 0.781+-0.012 | 0.070+-0.013 |
|      | pseudo_m | 0.511+-0.019 | 0.450+-0.018 | 0.779+-0.013 | 0.061+-0.008 |
|      | soft_reg | 0.493+-0.017 | 0.461+-0.015 | 0.766+-0.014 | 0.032+-0.004 |
|      | soft_reg_m | 0.499+-0.013 | 0.459+-0.011 | 0.768+-0.010 | 0.040+-0.005 |
|      | soft_reg_b | 0.497+-0.012 | 0.455+-0.010 | 0.769+-0.009 | 0.041+-0.005 |
| 0.10 | vanilla | 0.626+-0.051 | 0.499+-0.045 | 0.780+-0.010 | 0.126+-0.022 |
|      | reg_gap | 0.539+-0.025 | 0.482+-0.025 | 0.765+-0.013 | 0.057+-0.007 |
|      | pseudo | 0.518+-0.023 | 0.448+-0.022 | 0.779+-0.013 | 0.070+-0.012 |
|      | pseudo_m | 0.512+-0.023 | 0.445+-0.022 | 0.779+-0.015 | 0.066+-0.009 |
|      | soft_reg | 0.504+-0.020 | 0.461+-0.019 | 0.766+-0.015 | 0.043+-0.006 |
|      | soft_reg_m | 0.501+-0.014 | 0.453+-0.012 | 0.771+-0.010 | 0.048+-0.004 |
|      | soft_reg_b | 0.503+-0.013 | 0.454+-0.011 | 0.770+-0.010 | 0.049+-0.005 |
| 0.05 | vanilla | 0.824+-0.075 | 0.625+-0.053 | 0.780+-0.011 | 0.199+-0.040 |
|      | reg_gap | 0.643+-0.051 | 0.569+-0.044 | 0.760+-0.016 | 0.074+-0.019 |
|      | pseudo | 0.513+-0.018 | 0.443+-0.016 | 0.781+-0.012 | 0.070+-0.006 |
|      | pseudo_m | 0.519+-0.019 | 0.453+-0.018 | 0.777+-0.014 | 0.067+-0.007 |
|      | soft_reg | 0.527+-0.014 | 0.456+-0.013 | 0.770+-0.010 | 0.071+-0.004 |
|      | soft_reg_m | 0.512+-0.014 | 0.454+-0.015 | 0.769+-0.010 | 0.058+-0.003 |
|      | soft_reg_b | 0.517+-0.017 | 0.457+-0.016 | 0.768+-0.008 | 0.059+-0.004 |

Table 2: **Results on adult dataset:** The Results on Adult dataset for different methods and percentage of labeled data $D_L$ across different proportions of labeled and unlabeled data. We provide the average and the standard deviation over 50 runs for the accuracy, cross-entropy, and equalized odds as well as the total loss evaluated in the hold-out dataset. The total loss is the weighted sum of equalized odds and cross-entropy loss which is also the objective of the optimization as in 3

