# OpenReview forum: "A Soft Labeling Approach for Fairness-aware Learning Under Partially Annotated Sensitive Attributes"
_ICLR.cc/2024/Conference — Submitted to ICLR 2024_

### Official Review · Reviewer_UWvQ · 2023-10-24

**Soundness:** 3 good
**Presentation:** 2 fair
**Contribution:** 2 fair
**Rating:** 5
**Confidence:** 5

**Summary:**

This paper presents a solution to the challenge of algorithmic fairness in AI models when only partially annotated sensitive attributes are available, a common real-world scenario. Particularly, it proposes a method that utilizes soft pseudo-labels derived from attribute classifier output probabilities to mitigate bias. Additionally, the proposed methodology takes into account the uncertainty in attribute classifiers to further enhance the model’s performance in terms of fairness. The introduced approach is evaluated on two datasets, namely Adult (tabular data) and CelebA (visual data).

**Strengths:**

The paper addresses a highly important problem in the field of AI fairness, providing a practical solution for mitigating bias with limited access to protected attribute labels, a constraint that often occurs in real-world applications. The proposed approach is simple and effective on multiple data modalities (i.e., tabular and visual data).

**Weaknesses:**

1. The related works section is rather brief and could benefit from a more comprehensive overview of the existing literature within this domain. It also lacks a clear depiction of the limitations of previous approaches, which the proposed method aims to address. In general, this section fails to provide a clear view of the relevant works.

2. Regarding the methodology, limited details are provided for the auxiliary models f_z and f_d) (see Questions/Requests for more details)

3. The evaluation of the proposed method is not sufficient. Only two datasets are employed (Adult and CelebA), the evaluation protocol does not allow for assessing the overall performance of the compared methods (i.e., the trade-off between accuracy and fairness), and several state-of-the-art approaches are not considered in the comparison. (see Questions/Requests for more details).

4. There are several typos in the manuscript that should be fixed.

**Questions:**

1. It is not clear how the auxiliary model trained on only 5% of the data can learn the desired information. Does the size of the entire dataset affect the performance of the method? Have the authors investigated the possibility of introducing other types of biases when training the auxiliary models on such limited data? Considering the crucial role of the auxiliary models in the proposed method, the authors should provide an analysis pertaining to models.

2. Figure 1 shows that the proposed approach is fairer but less accurate compared to the baselines. Is the overall performance (i.e., considering both utility and fairness) of the proposed method higher than the performance of the competitive approaches? How does the hyperparameter \lambda affect the results?

3. Regarding the visual data, there are widely applied benchmarks (see [1]), such as Colored-MNIST, UTKFACE, and CelebA (with isBlonde and HeavyMakeup targets) for assessing fairness-aware approaches. In order to provide a comprehensive evaluation of the proposed method, the authors should follow this evaluation setup and consider the state-of-the-art approaches in the comparison [1,2,3].

4. How does the proposed method perform under different levels of bias in the training data?

5. How does the proposed method perform for multiclass protected attributes (e.g., race)?

[1] Hong, Y., & Yang, E. (2021). Unbiased classification through bias-contrastive and bias-balanced learning. Advances in Neural Information Processing Systems, 34, 26449-26461.
[2] Barbano, C. A., Dufumier, B., Tartaglione, E., Grangetto, M., & Gori, P. (2022). Unbiased supervised contrastive learning. arXiv preprint arXiv:2211.05568.
[3] Sarridis, I., Koutlis, C., Papadopoulos, S., & Diou, C. (2023). FLAC: Fairness-Aware Representation Learning by Suppressing Attribute-Class Associations. arXiv preprint arXiv:2304.14252.

---

### Official Review · Reviewer_dNWk · 2023-10-30

**Soundness:** 3 good
**Presentation:** 2 fair
**Contribution:** 2 fair
**Rating:** 3
**Confidence:** 2

**Summary:**

This paper proposes a fairness-aware regularization method that makes use of the soft labels of attribute classifiers to alleviate the information of a group-unlabeled dataset. In addition, the effect of the uncertainty of the attribute classifier is also taken into consideration. This method can be used to learn fairness-aware models when only partial information about sensitive attributes is available.

**Strengths:**

This paper studies the problem of fairness-aware learning under partially sensitive attribute information. Different from conventional methods which rely on an attribute classifier as a proxy model to infer "hard" pseudo labels, this paper proposes a soft labeling method with a novel perspective. The algorithm part is described in detail and logically clear.

**Weaknesses:**

1. The presentation of the paper should be improved. The motivation of the paper in the introduction is not clear. The introduction part is not well organized.
2. The datasets used in the experiment were too small to be convincing.
3. The number of comparison algorithms in the experimental part is small.
4. Some of the references cited in the paper are outdated or not directly relevant to the proposed method.

**Questions:**

1. The setup of this method is based on binary classification. Can it be extended to multi-classification?
2. This paper emphasizes the study of the problem of fairness-aware learning under partially sensitive attribute information. Could you give a specific definition of “partially”?
3. For the experimental results, in terms of fairness, the soft regularization method in this paper consistently achieves lower bias across varying proportions of labeled and unlabeled samples. It is beneficial to analyze the reasons for the bias.

---

### Official Review · Reviewer_jukW · 2023-11-01

**Soundness:** 2 fair
**Presentation:** 2 fair
**Contribution:** 2 fair
**Rating:** 3
**Confidence:** 4

**Summary:**

This paper addresses the challenge of algorithmic fairness when sensitive attributes are partially known. Traditional methods use an attribute classifier to generate "hard" pseudo labels for fairness-aware model optimization. The authors propose a new regularization method that employs "soft" pseudo labels based on the attribute classifier's output probabilities, aligned with fairness criteria definitions. They also explore how uncertainty in attribute classifier parameters—due to limited sensitive attribute data—affects outcomes. Experiments on the Adult and CelebA datasets show that their method outperforms existing baselines in terms of utility and fairness, demonstrating the importance of considering uncertainty.

**Strengths:**

1. The paper proposes to use soft pseudo labels to create a new fair regularization term for fair machine learning. The idea is simple yet efficient.
2. Experiments are conducted on both tabular and image datasets.

**Weaknesses:**

1. Using a proxy model trained on labeled sensitive attributes to predict sensitive attributes is not new [R1]. Using a proxy model to obtain soft sensitive attributes is also not new [R2]. Although these works mainly focus on evaluating fairness, they can be directly extended to learning fair models with standard fair training algorithms.
2. The sensitive attributes predicted by proxy models are biased [R3], so the soundness of this approach needs more discussion.
3. Only "Equalized Odds" is considered, which is insufficient.
4. The numbers in Table 1 may not be convincing. Even with 5% labeled data, the task smiling can still achieve an accuracy of 85% with an Equalized Odds of 0.05.

[R1] Evaluating fairness of machine learning models under uncertain and incomplete information. FAccT 2021.

[R2] Fairness under unawareness: Assessing disparity when protected class is unobserved. FAT 2019.

[R3] Weak Proxies are Sufficient and Preferable for Fairness with Missing Sensitive Attributes. ICML 2023.

**Questions:**

Please address the weakness above.

---

### Author Response · Authors · 2023-11-23

Dear Reviews,

We sincerely appreciate the time and effort you invested in reviewing our work.

Your thoughtful questions and insights have been incredibly valuable, helping us identify areas in our paper that need further clarification or elaboration.  Unfortunately, addressing each specific question would require substantial revisions to the manuscript. However, we are fully committed to incorporating your suggestions to enhance the overall quality of our future submissions.

Thank you for your understanding and support, your feedback is immensely valuable to us.

---

### Meta-Review · Area_Chair_BmgX · 2023-12-09

**Metareview:**

This paper tackles the issue of algorithmic fairness in scenarios where only partial information about sensitive attributes is available. Conventional methods rely on "hard" pseudo labels generated by an attribute classifier for fairness-aware model optimization. In contrast, the authors introduce a novel regularization technique that utilizes "soft" pseudo labels based on the output probabilities of the attribute classifier, aligning with fairness criteria definitions. The paper also investigates the impact of uncertainty in attribute classifier parameters, arising from limited sensitive attribute data. Experimental results on the Adult and CelebA datasets demonstrate the superiority of their method over existing baselines in terms of both utility and fairness, underscoring the significance of accounting for uncertainty in addressing algorithmic fairness challenges.

The reviewers raised several questions regarding the proposed method’s merits in light of the literature of handling imperfect sensitive attributes. The reviewers also believe the paper can benefit from adding more clarifications to the proposed approach and offers a stronger set of experiments and comparisons.

**Justification For Why Not Higher Score:**

The paper should better clarify its merits wrt a line of highly relevant literature that adopted similar design principles. More substantial experiments and comparisons are encouraged.

**Justification For Why Not Lower Score:**

N/A

---

### Decision · Program_Chairs · 2024-01-16

Reject